# Study of the Hazard of Endogenous Fires in Coal Mines—A Chemometric Approach

**Karolina Wojtacha-Rychter [1] and Adam Smoliński [2],\*** 

[1]   Central Mining Institute, Department of Mining Aerology, Pl. Gwarków 1, 40-166 Katowice, Poland; kwojtacha@gig.eu
[2]   Central Mining Institute, Pl. Gwarków 1, 40-166 Katowice, Poland
\*   Correspondence: smolin@gig.katowice.pl; Tel.: +48-32-259-2252

**Abstract:** The most commonly used practice to assess fire hazard development in underground coal mines is based on the measurement of the concentration of selected gases in the mine's air. The main goal of this study was present a strategy to monitor the gaseous atmosphere in the mine in order to identify the onset of an endogenous fire in the coal seam. For that purpose, the principal component analysis (PCA) and the hierarchical clustering analysis (HCA) were applied. The monitoring covers the measurements of concentration of $CO$, $CO_2$, $H_2$, $O_2$, $N_2$, and selected hydrocarbons, respectively throughout the whole of one year. The chemometric methods applied allow for effective exploration of the similarities between the studied samples collected both under fire hazard conditions and under safe conditions. Based on the constructed models, the groups of objects characterized with the highest content of ethylene, acetylene, propylene, and carbon monoxide were identified. These samples indicate the endogenic fire in coal mine.

**Keywords:** coal; self-heating; mine fire; hierarchical clustering analysis; principal component analysis

## 1. Introduction

All over the world, coal constitutes one of the three primary energy sources used for the generation of electricity and commercial heat [1,2]. Such a large use of coal is determined by the sufficiency and availability of its resources, the ease and safety of transport, and the low cost of coal-based energy production. However, underground mining still remains an industry of high safety hazard due to difficult mining and geological conditions as well as the occurrence of natural threats such as: methane and coal dust explosions, endogenous fires, collapsing of mine stopes, and mining-induced seismicity or flooding. In underground mines, fire hazard is the cause of the most dangerous and the most frequent accidents. The high levels of deep ground stress and ground temperature provide favorable conditions for coal self-heating, thereby increasing the risk of fire. The self-heating process is initiated by the increase of coal temperatures resulting from highly exothermic oxidation reactions during the exposure of coal mass to oxygen from mine air [3–6]. If the heat produced during the oxidation reaction accumulates, the temperature starts to increase up to the ignition point. After that, if favorable conditions continue, self-ignition and fire take place. The detection of the coal self-heating process in its early stage is very important to prevent fires and to more easily deal with them. A mine fire may be detected by physical symptoms recognized by means of human senses. The physical symptoms include the following: odor caused by the emission of aromatic hydrocarbons, formation of water drops on the roofs, walls, or any other cooler surface, and sound originating from the expansion, cracking, and falling of strata due to the advancement of fire. The fire may be also detected by thermal devices and by chemical methods [7,8]. As far as the chemical method is concerned, it is based on the monitoring of fire gas concentrations and then analyzing the changes in air composition.

The coal and oxygen reaction results from the release of heat as well as the emission of gaseous products, typically carbon monoxide, carbon dioxide, hydrogen, as well as saturated and unsaturated hydrocarbons. There are several papers published which discuss the monitoring of gases emitted from the source of coal heating [9–14]. In the course of the said studies, it was mainly observed that the concentration of the gases increases with the temperature increase, which was applied to determine the degree of fire hazard in coal mines. Carbon monoxide constitutes the main indicator due to the fact that it is formed in large amounts and, thus, it is easy to detect at lower temperatures in comparison to other gases [14,15]. But, the excess emission of this gas caused by some other phenomena apart from the self-heating process may limit the application of carbon monoxide as a fire indicator [16]. Hydrocarbons, particularly such alkenes as ethylene, propylene, and acetylene, are also emitted as the temperature rises. However, the low level of hydrocarbon concentration in mine air and their detection at temperatures higher than carbon monoxide are the reasons why these gases are less sensitive indicators in the early stages of coal oxidation [12,17]. The results of quantitative analyses of single gaseous components are also used to construct the so-called fire indices such as: Graham's ratio, Young's ratio, Willet's ratio, $CO/CO_2$ ratio, and Morris ratio [15,18]. Despite some limitations, the widely used method of fire hazard assessment that was proposed and studied involves collecting samples of mine air into airtight bags and then detecting the presence of coal thermal decomposition and oxidation products with the application of a gas chromatograph [19–21]. Data obtained in the long-term period illustrate the trend of changes in the concentrations of these gases in the selected coal seam. The data of long-term controlling of the composition of mine air are usually very complex due to a number of analyzed parameters and time variable. A graphic presentation of the collected data as the changes in each gas concentration or fire indices in time constitutes a commonly used method to visualize the status of mine air [22]. The results presented in the literature usually originate from the simulation of coal fire in laboratory conditions. The number of the results obtained in the experiments is lower than the number of the results achieved in real conditions underground; in addition, the relationship among the data is clearly visible [14,23]. The presentation of the results from long-term measurements and for all gases on separate curves can make the interpretation of the relationship among all gases difficult. The idea of employing chemometric methods in the development of measurement data available from monitoring the air state in the underground coal mines constituted the fundamental novelty of this publication. In this paper, a strategy enabling the investigation of the monitoring of a gaseous atmosphere in the coal mine to identify the onset of endogenous fire in the coal seam was proposed.

## 2. Materials and Methods

Principal component analysis (PCA) [24–29] is a method of data exploration and visualization. It transfers studied parameters which are possibly correlated into new linearly uncorrelated variables called principal components (PCs). Mathematically, the studied data organized in data matrix $\mathbf{X}(m, n)$ are decomposed into two matrices $\mathbf{S}(m, f)$ and $\mathbf{D}(n, f)$, called score and loading matrix, respectively:

$$\mathbf{X}(m \times n) = \mathbf{S}(m \times f) \bullet \mathbf{D}'(n \times f) + \mathbf{R}(m \times n) \tag{1}$$

where $m$ and $n$ denote objects (studied samples) and parameters, respectively, whereas $f$ denotes the number of significant PCs and $\mathbf{R}$ is a residuals matrix. The following PCs are constructed to maximize the description of the data variance.

In the case when data compression is ineffective, the alternative exploration method, called Hierarchical Clustering Analysis (HCA) [30–34] can be applied. It allows investigating the similarities/dissimilarities between studied samples (objects) in the variables space and variables (measured parameters) in the object space. There exist various options of HCA differing in the similarity measures used as well as the way of merging similar objects. The Euclidean distance is the similarity measure most commonly applied, whereas, among the linkage methods, the most often

used one is the Ward linkage method based on the inner squared distance of clusters. The results of HCA are shown as dendrograms. One dendrogram presents the similarities between studied objects (samples) in the parameters space and the other one shows the similarities between studied variables in the objects space. Due to the fact that classical HCA does not allow investigating the similarity between studied samples in the parameters space and the similarity between studied variables in the objects space simultaneously, a color map of the data was proposed [35]. The color map presents the original data in the form of color pixels sorted according to the order presented in HCA dendrograms.

The data set of a long-term monitoring of endogenous fire threat in a hard coal mine was organized in matrix **X** (180 × 33) whose rows represent days when samples were collected, whereas the columns represent 11 studied parameters (see Table 1) measured in three places: The route, and behind the dam at a distance of 20 m and 350 m, respectively. All studied samples were collected for one year. Objects nos. 1–9 were collected in January, objects nos. 10–27 in February, objects nos. 28–54 in March, objects nos. 55–76 in April, objects nos. 77–94 in May, objects nos. 95–107 in June, objects nos. 108–119 in July, objects nos. 120–129 in August, objects nos. 130–141 in September, objects nos. 142–156 in October, objects nos. 157–167 in November, and objects nos. 168–180 in December 2017. The studied data set applied in PCA and HCA analyses was standardized:

$$x_{ij} = \frac{(x_{ij} - \overline{x}_j)}{s_j} \tag{2}$$

where $\overline{x}_j, s_j$ is the mean of the *j*-th column and its standard deviation, respectively:

$$x_j = \frac{1}{m} \sum_{i=1}^{m} x_{ij} \tag{3}$$

$$s_j = \sqrt{\frac{1}{m} \sum_{i=1}^{m} (x_{ij} - x_j)^2} \tag{4}$$

whereas m denotes the number of objects.

A gas sampling bag, also known as Tedlar bag, was used to collect air samples. The analysis of the quality of gaseous components was made with the application of gas chromatography. A flame ionization detection was applied to determine carbon dioxide, saturated hydrocarbon (ethane and propane), and unsaturated hydrocarbons (ethylene, propylene and acetylene). Whereas, for determining hydrogen, carbon monoxide and carbon dioxide, methane, oxygen, and nitrogen, a thermal conductivity detector (TCD) was used. Gas samples were injected into the chromatographic system via a loop column. The high purity argon was used as a carrier gas for hydrogen but in cases of the rest of the gases a pure helium was chosen. For the purpose of obtaining optimal selectivity and sensitivity, suitable intensities of gas stream flow and the temperature of the column were set. The low temperature of the column was chosen for hydrogen −303 K, and for carbon dioxide, methane, oxygen, and nitrogen −318 K. The column temperature for carbon dioxide was the highest and equal to 975 K. For hydrocarbons a column temperature program varied from the initial oven temperature 323 K to the temperature 472 K. Measurement uncertainties for each gas were calculated according to equations, which were determined on the basis of measurements using the certified reference material in an accredited laboratory. The value of uncertainty of the gases' concentration varied depending on the gas amount. For example, hydrocarbons had an measurement error of 10 ppm (for gas concentration 100 ppm) and 1 ppm (for gas concentration 10 ppm). Hydrocarbons had a measurement error of 10 ppm (for gas concentration >100 ppm) and 1 ppm (for gas concentration >0.01 ppm). Acetylene had a measurement error of 0.002 ppm (for gas concentration <0.01 ppm) and 0.012 ppm (for gas concentration >0.01 ppm). Hydrogen had a measurement error of 1 ppm (for gas concentration 10 ppm), but for the highest hydrogen concentration the error was equal to 300 ppm.

The oxygen, nitrogen, methane, and carbon dioxide had measurement errors of 0.5%, 2%, 1.5%, 0.2%, respectively, independently from gas concentration.

**Table 1.** The measured parameters.

| No. | Parameters, ppm | Sample Collection Location |
|---|---|---|
| 1 | concentration of ethane | |
| 2 | concentration of ethylene | |
| 3 | concentration of propane | |
| 4 | concentration of propylene | |
| 5 | concentration of acetylene | |
| 6 | concentration of carbon monoxide | measured in the route |
| 7 | concentration of oxygen | |
| 8 | concentration of nitrogen | |
| 9 | concentration of carbon dioxide | |
| 10 | concentration of methane | |
| 11 | concentration of hydrogen | |
| 12 | concentration of ethane | |
| 13 | concentration of ethylene | |
| 14 | concentration of propane | |
| 15 | concentration of propylene | |
| 16 | concentration of acetylene | |
| 17 | concentration of carbon monoxide | measured behind the dam at a distance of 20 m |
| 18 | concentration of oxygen | |
| 19 | concentration of nitrogen | |
| 20 | concentration of carbon dioxide | |
| 21 | concentration of methane | |
| 22 | concentration of hydrogen | |
| 23 | concentration of ethane | |
| 24 | concentration of ethylene | |
| 25 | concentration of propane | |
| 26 | concentration of propylene | |
| 27 | concentration of acetylene | |
| 28 | concentration of carbon monoxide | measured behind the dam at a distance of 350 m |
| 29 | concentration of oxygen | |
| 30 | concentration of nitrogen | |
| 31 | concentration of carbon dioxide | |
| 32 | concentration of methane | |
| 33 | concentration of hydrogen | |

## 3. Results

PCA and HCA methods were applied to identify the beginning of endogenous fires in the coal mine. The classical PCA did not allow effective reduction of data dimensionality; therefore, only very general conclusions on the studied phenomenon could be presented based on its results. The PCA model with 5 PCs describes 83.84% of data variance. Score plots and loading plots are shown in Figure 1.

Along PC1 (presenting 47.41% of the total data variance) the studied objects may be grouped into three main clusters. The first cluster includes the majority of objects (nos. 1–11, 23–36 and 39–180), the second cluster collects objects nos. 12–17, 20–22, 37, and 38, whereas the third cluster groups only two objects (nos. 18 and 19). The loading plot PC1 vs. PC2 enables us to explain the differences between objects nos. 12–22, 37, 38, and all the remaining objects resulting from relatively higher concentrations of ethane, ethylene, propane, propylene, acetylene, carbon monoxide, and methane were measured behind the dam at a distance of 20 m (parameters nos. 12–17, 21). Ethane, propane, propylene, acetylene, carbon monoxide, and hydrogen were measured behind the dam at a distance of 350 m (parameters nos. 23, 25–28 and 33). Moreover, objects nos. 18 and 19 characterized with the highest concentration of ethylene, acetylene, and carbon monoxide were measured in the route; ethane,

ethylene, propane, propylene, and acetylene were measured behind the dam at a distance of 20 m (parameters nos. 12–16). Ethane, propane, propylene, acetylene, and carbon monoxide were measured behind the dam at a distance of 350 m (parameters nos. 23, 25–28).

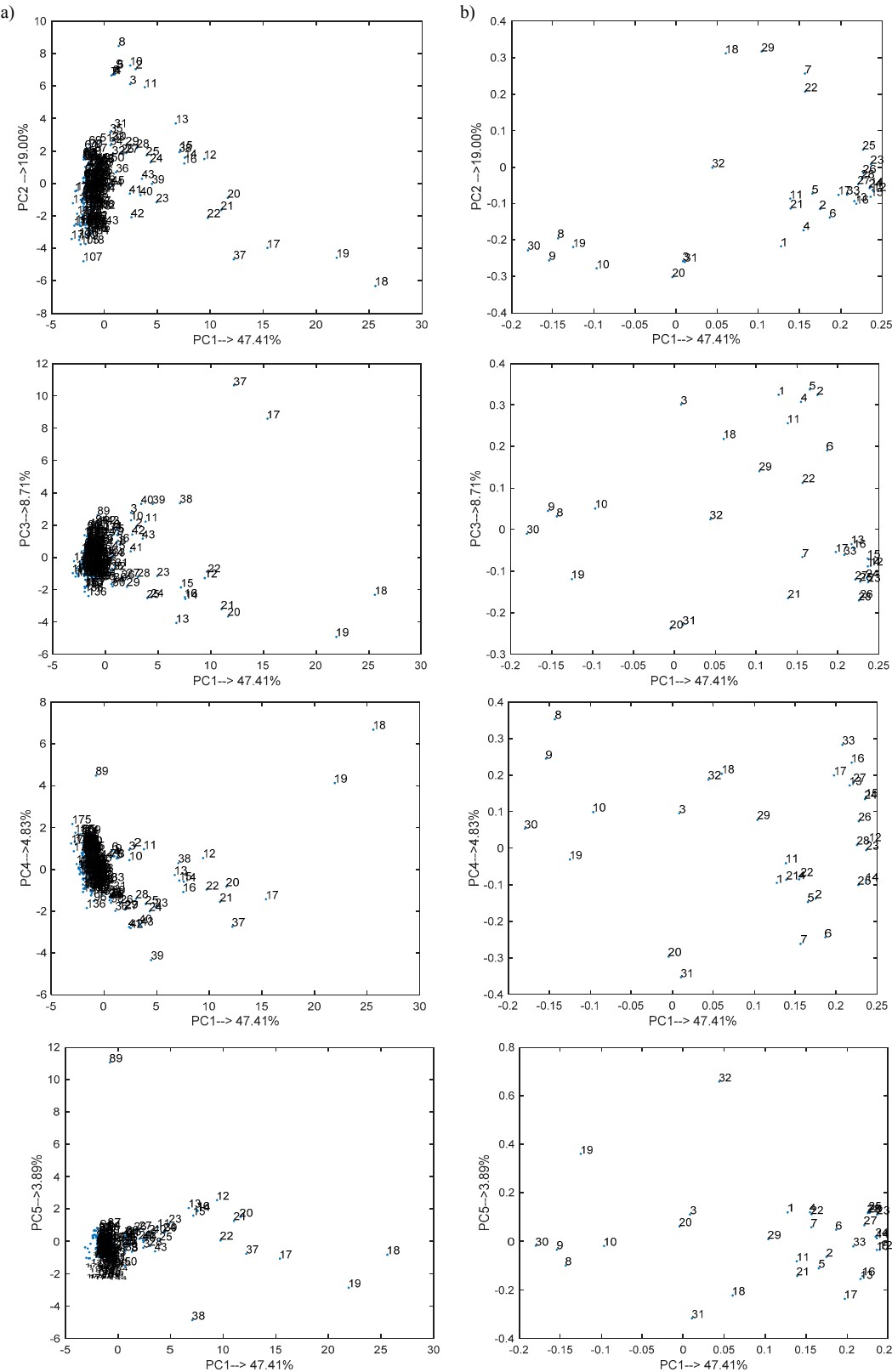

**Figure 1.** (**a**) Score plots and (**b**) loading plots as a result of PCA for the standardized data **X** (180 × 33).

Additionally, PC2 which describes 19.00% of the data variance shows the difference between objects nos. 17–19 and 37 and objects nos. 2–11. Objects nos. 2–11 characterized with high concentrations of oxygen were measured in the route and hydrogen was measured behind the dam at a distance of 20 m (parameters nos. 7 and 22) and the highest concentration of oxygen was measured behind the dam at a distance of 20 m and 350 m (parameters nos. 18 and 29, respectively) among all the studied objects. Objects nos. 17–19 and 37 characterized with high concentration of carbon dioxide were measured behind the dam at a distance of 20 m (parameter no. 20) and low concentrations of oxygen were measured behind the dam at a distance of 20 m and 350 m (parameters nos. 18 and 29, respectively) in comparison with all the remaining objects.

PC3, which describes 8.71% of the data variance, represents the difference between objects nos. 17 and 37 and all the remaining objects due to the relatively high concentration of ethane, ethylene, propylene, and acetylene were measured in the route (parameters nos. 1, 2, 4 and 5) and low concentrations of carbon dioxide were measured behind the dam at a distance of 20 m and 350 m (parameters nos. 20 and 31, respectively).

The PC4, describing 4.83% of the total variance, reveals the uniqueness of objects nos. 18 and 39, whereas the PC5, describing 3.89% of the total variance proves the uniqueness of object no. 89. Object no. 18 characterized with the highest concentration of nitrogen (parameter no. 8) and the lowest concentration of carbon dioxide was measured behind the dam at a distance of 20 m (parameter no. 20) among all the studied samples. Objects no. 89 is unique due to the highest concentration of methane and the lowest concentration of carbon dioxide and was measured behind the dam at a distance of 350 m (parameters nos. 32 and 31, respectively).

Since the data compression in PCA was ineffective, and enabled drawing only general conclusions on the phenomenon studied, HCA method was applied to effectively explore the internal data structure and investigate the similarities and differences between studied objects in the parameters space and between parameters in the objects space, respectively. The results of HCA are presented in Figure 2.

On the basis of the dendrogram presented in Figure 2a three main clusters may be distinguished:

- Cluster A collecting objects nos. 1–16, 20–35, and 38–43;
- Cluster B composed of objects nos. 17–19 and 37;
- Cluster C grouping objects nos. 36 and 44–180.

Moreover, within cluster A three sub-clusters can be observed: Sub-cluster A1 collecting objects nos. 1–11, sub-cluster A2 grouping objects nos. 12–16, and 20–22 and sub-cluster A3 composed of objects nos. 23–35 and 38–43, respectively.

The dendrogram presented in Figure 2b groups the studied parameters into four main classes:

- Class A: Parameters nos. 12–17, 21, 23–28 and 33 (describing concentrations of ethane, ethylene, propane, propylene, acetylene, and carbon monoxide, measured behind the dam at a distance of 20 m; concentration of methane was measured behind the dam at a distance of 20 m, as well as concentrations of ethane, ethylene, propane, propylene, acetylene, carbon monoxide, and hydrogen, measured behind the dam at a distance of 350 m, respectively);
- Class B: Parameters nos. 7, 18, 22 and 29 (describing concentration of oxygen measured in the route, concentrations of oxygen and hydrogen measured behind the dam at a distance of 20 m, and concentration of oxygen measured behind the dam at a distance of 350 m);
- Class C: Parameters nos. 1–6, 11 and 32 (concentrations of ethane, ethylene, propane, propylene, acetylene, carbon monoxide, and hydrogen measured in the route, as well as concentration of methane measured behind the dam at a distance of 350 m); and
- Class D: Parameters nos. 8–10, 19, 20, 30 and 31 (describing concentrations of nitrogen, carbon dioxide, and methane measured in the route, as well as concentrations of nitrogen and carbon dioxide measured behind the dam at a distance of 20 m and 250 m, respectively).

The color map (Figure 2c) presents the studied parameters in the form of a color image with pixels representing the data matrix elements sorted according to the order given in Figure 2a,b. On the basis

of the analysis of the dendrograms complemented with the color map of the studied data, it may be concluded that the objects collected in cluster A showed a relatively higher concentration of oxygen measured in the route and behind the dam at a distance of 20 m (parameters nos. 7 and 18); a higher concentration of hydrogen was measured behind the dam at a distance of 20 m (parameter no. 22) and a higher concentration of oxygen was measured behind the dam at a distance of 350 m (parameter no. 29) than all the remaining objects.

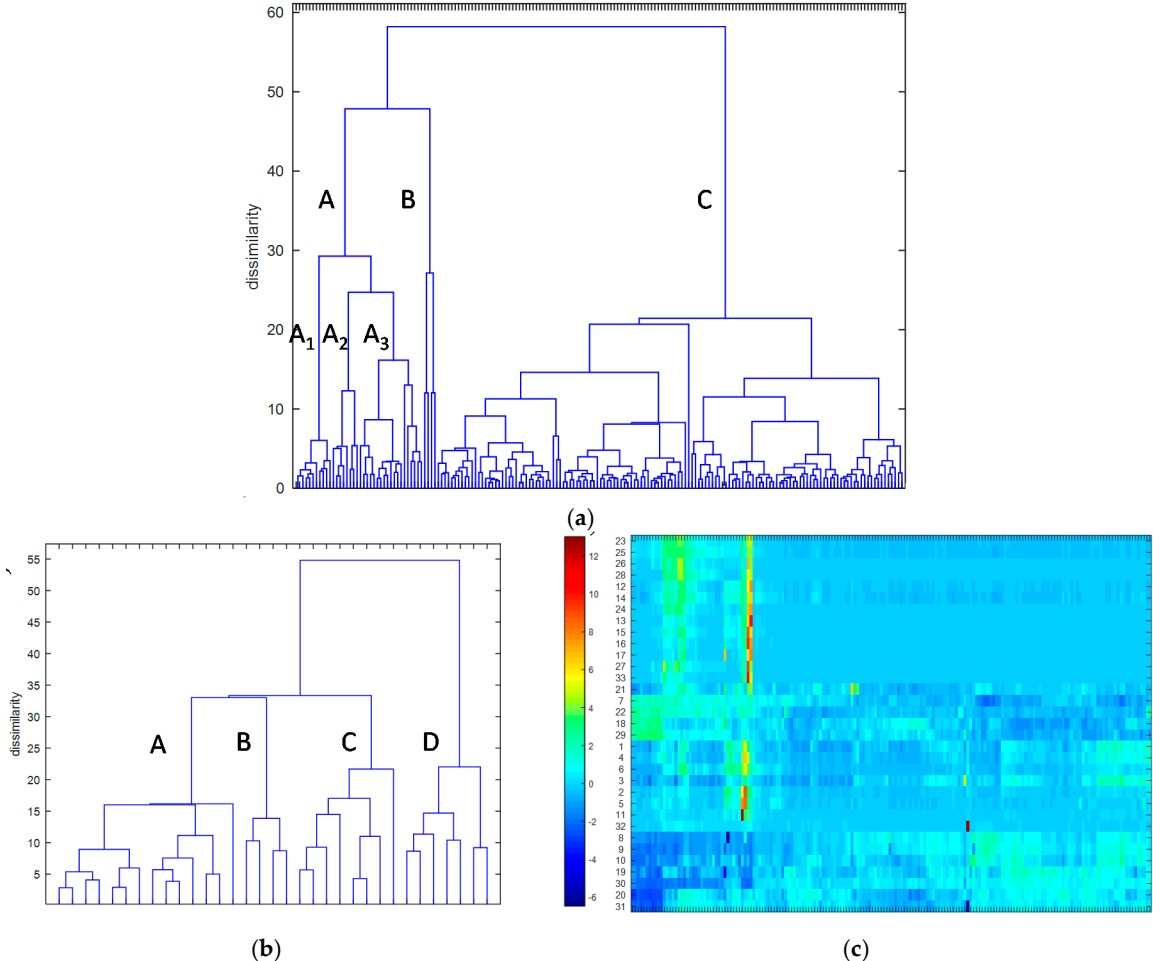

**Figure 2.** Dendrograms of (**a**) objects (days when samples were collected) in the space of 33 parameters (listed in Table 1) and (**b**) variables in the space of 180 objects with (**c**) a color map of the studied data sorted according to the Ward linkage method.

Moreover, the color map also shows that objects nos. 1–11 (grouped in sub-cluster A1) characterized with the highest concentration of oxygen measured in the route and behind the dam at a distance of 20 m (parameters nos. 7 and 18), a concentration of hydrogen measured behind the dam at a distance of 20 m (parameter no. 22), and a concentration of oxygen measured behind the dam at a distance of 350 m (parameter no. 29); as well as the lowest concentration of carbon dioxide and methane measured in the route (parameters nos. 9 and 10), concentration of carbon dioxide measured behind the dam at a distance of 20 m (parameter no. 20), and concentrations of nitrogen and carbon dioxide measured behind the dam at a distance of 350 m (parameters nos. 30 and 31), among all the studied samples.

Objects nos. 12–16 and 20–22 collected in sub-cluster A2 are unique due to high concentrations of ethane, propylene, carbon monoxide, and oxygen measured in the route (parameters nos. 1, 4, 6 and 7); concentrations of ethane, ethylene, propane, propylene, and acetylene measured behind the dam at a distance of 20 m (parameters nos. 12–17); concentrations of methane and hydrogen measured behind

the dam at a distance of 20 m (parameters nos. 21, 22); and concentrations of ethane, ethylene, propane, propylene, acetylene, carbon monoxide, oxygen, and hydrogen (parameters nos. 23–29 and 33).

Sub-cluster A3 collecting objects nos. 23–35 and 38–43 characterized with the lowest concentrations of ethane, propane, propylene, acetylene, carbon monoxide, and hydrogen were measured behind the dam at a distance of 350 m (parameters nos. 23, 25–28 and 33), and the highest concentration of carbon dioxide was measured behind the dam at a distance of 20 m and 350 m (parameters nos. 20 and 31), among all the objects belonging to cluster A.

Cluster B is unique due to the high concentration of oxygen measured in the route (parameter no.7) and the concentrations of methane and hydrogen measured behind the dam at a distance of 20 m (parameters nos. 21 and 22). Low concentrations of nitrogen, carbon dioxide, and methane were measured in the route (parameters nos. 8–10), and a concentration of nitrogen was measured behind the dam at a distance of 350 m (parameter no. 30). The value of parameters nos. 8–10, and 30, and the highest concentrations of ethane, ethylene, propylene, acetylene, carbon monoxide, and hydrogen were measured in the route (parameters nos. 1, 2, 4–6, 11). Concentrations of ethane, ethylene, propane, propylene, acetylene, and carbon monoxide measured behind the dam at a distance of 20 m and 350 m (parameters nos. 12–17 and 23–28, respectively), and the concentration of hydrogen measured behind the dam at a distance of 350 m (parameter no. 33), among all the studied objects.

Cluster C grouping objects nos. 36 and 44–180 characterized with relatively higher concentrations of nitrogen, carbon dioxide, and methane were measured in the route (parameters nos. 8–10), and a concentration of nitrogen was measured behind the dam at a distance of 20 m (parameter no. 19), as well as a concentration of hydrogen that was measured behind the dam at a distance of 350 m (parameter no. 30).

## 4. Conclusions

An effective way of identifying fire hazards occurring in a real coal mine was presented. The applied chemometric methods allows us to explore studied mine air monitoring data in terms of fire hazard assessment. PCA allowed distinguishing the samples corresponding to the various conditions in the underground coal mine, whereas HCA additionally divided the studied monitoring period into three clusters. Cluster A, separated in both PCA and HCA, collected monitoring gas samples with high concentrations of oxygen, which reflects safe conditions in coal mine. In the PCA results, Cluster B grouped monitoring data samples with higher contents of ethane, ethylene, propane, propylene, acetylene, carbon monoxide, and methane than in cluster A. The increase of the content of these gases may suggest the beginning of coal self-heating. Cluster C included monitoring gas samples characterized with the highest contents of ethylene, acetylene, and carbon monoxide, which indicated the occurrence of fire. In HCA analysis, cluster B probably reflects the fire state because of groupings of some objects characterized with the highest concentrations of the above gases measured behind the dams and in the route. The carbon monoxide, ethylene, acetylene, propylene, and carbon monoxide are the gases most commonly used to assess the phenomenon of coal self-heating, and thereby the fire hazard as well, because with the increase of coal temperature the characteristic emission of these gases occurs. The concentrated objects in cluster C in the Hierarchical Cluster Analysis differed from the objects concentrated in clusters A and B in terms of higher concentrations of nitrogen and carbon dioxide. Nitrogen and carbon dioxide are the gases most commonly used for inertisation of mine air to considerably reduce oxygen concentration in goaf air. Thus, Cluster C may reflect the conditions of fire prevention activities or firefighting operations. Based on the results, it can be concluded that the applied chemometric methods can have a great application potential in the assessment of fire hazard in underground coal mines. The approach allows identifying the samples with gases' compositions clearly different from other samples taken from the same place but at different time intervals. These samples may indicate that a low-temperature oxidation process of coal starts, which finally leads to self-ignition and fire in the coal mine. However, the success of this method depends on the quality of the analysis and quick detection of fire gases in the mine atmosphere, particularly such gases as hydrocarbons

whose concentration at the beginning of the process is very low. In the course of the experiment, the range of analyzed parameters in chemometric methods can be extended by means of additional parameters characterizing the physico-chemical properties of the coal deposit or the place where the gaseous samples are collected. This conception indicates the direction of the future research in this area. The above method provides an opportunity to compress the information in many variables by using a small number of principal components. It seems very useful to extend the range of the parameters such as petrographic components, ash, and moisture content or the depth of coal deposits because they have a great impact on the development of the self-heating process of coal.

**Author Contributions:** K.W.-R. and A.S. conceived and designed of the study; K.W.-R. collected the data; A.S. analyzed the data, K.W.-R.; and A.S. interpreted and wrote the paper.

**Funding:** This work was supported by the Ministry of Science and Higher Education, Poland grant number [14205027].

**Conflicts of Interest:** The authors declare no conflict of interest. The founding sponsors had no role in the design of the study; in the collection, analyses, or interpretation of data; in the writing of the manuscript, and in the decision to publish the results.

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
