# Peer review of "Study of the Hazard of Endogenous Fires in Coal Mines—A Chemometric Approach"

_energies, doi:10.3390/en11113047_

Round 1

Reviewer 1 Report

This study developed a general strategy allowing the monitoring of gaseous atmosphere in the mine in order to identify the onset of an endogenous fire in the coal seam. Both he principal component analysis (PCA) and the hierarchical clustering analysis (HCA) were applied to analyse the sampling data. The monitoring covers the measurements of concentration of carbon monoxide, carbon dioxide, hydrogen, oxygen, nitrogen and selected hydrocarbons, respectively throughout the whole one year. The chemometric methods applied allow to effectively explore the similarities and dissimilarities between the studied samples collected both under fire hazard conditions and under safe conditions. Overall, this work is well-written and the argument is pretty clear. Some minor revisions are needed before it is accepted. 

One more sections are needed to clarify how the measured data in Table 1 is obtained, which should include measuring instruments, sampling techniques and data error analysis.  

The authors are recommended to add one more section to discuss the limitation of the proposed method. 

Figure 2 needs to be improved for clarification. 

Author Response

       I.            Reviewer #2:

This study developed a general strategy allowing the monitoring of gaseous atmosphere in the mine in order to identify the onset of an endogenous fire in the coal seam. Both he principal component analysis (PCA) and the hierarchical clustering analysis (HCA) were applied to analyse the sampling data. The monitoring covers the measurements of concentration of carbon monoxide, carbon dioxide, hydrogen, oxygen, nitrogen and selected hydrocarbons, respectively throughout the whole one year. The chemometric methods applied allow to effectively explore the similarities and dissimilarities between the studied samples collected both under fire hazard conditions and under safe conditions. Overall, this work is well-written and the argument is pretty clear. Some minor revisions are needed before it is accepted.

1.      One more sections are needed to clarify how the measured data in Table 1 is obtained, which should include measuring instruments, sampling techniques and data error analysis. 

Response: The section numbered 2 was completed with a discussion on measurement the gas concentrations (Please see lines: 117 – 130).

2.      The authors are recommended to add one more section to discuss the limitation of the proposed method.

Response: Indeed the limitation of the PCA and HCA methods used in the exploration of the studied experimental data were presented in the text (see lines 176-180).  Namely PCA could be (and was in our study) ineffective when the studied parameters are independent (orthogonal) or when some non-linear correlation between studied parameters are observed. To overcome this disadvantage the HCA was used. 

3.      Figure 2 needs to be improved for clarification.

Response: Shortly the idea of the color map  was explained in line 210 and 211. The dendrogram reveals data structure (i.e. the sub-groups of objects), but it allows no interpretation of the observed patterns in terms of the original variables (parameters). For this purpose, we used a simple visualization method, the principle of which can be presented as follows: Let us assume that the studied data set is organized in the matrix form containing m objects and n variables, X (m ´ n). If hierarchical clustering is applied to data objects, then along axis x of the resulting dendrogram, there are m ordered objects. Let us denote this specific order of objects as the 'objorder'. One simple way of interpretation of the resulting clustering tree would be to display the data set with objects sorted according to the 'objorder' as an image, with pixels representing the matrix elements. However, if the measured parameters are independent variables, then their random (arbitrary) order in the data matrix introduces abrupt disturbances in the image. To overcome this problem, we propose to sort data matrix in the variables' direction as well. The order of the variables can be estimated, based on the results of their clustering in an analogous way, as the 'objorder' was calculated. The order of variables will in the subsequent parts of this text be denoted as 'varorder'. Then the resulting image of the data set attains a smoother appearance, because the neighboring objects and variables are ordered according to their similarity.

Reviewer 2 Report

This is surely an interesting and original work. It has the potential to be published in Energies. I have only a couple of comments that the authors should implement into the revised manuscript prior to publication.

1) Introduction - The aim of the work and its connection with the literature gaps should be more extensively described.

2) Conclusion - The practical impact of the results obtained in this work should be better highlighted. The authors should also give an outlook on future research work.

Author Response

Reviewer #3: This is surely an interesting and original work. It has the potential to be published in Energies. I have only a couple of comments that the authors should implement into the revised manuscript prior to publication.

1.      Introduction - The aim of the work and its connection with the literature gaps should be more extensively described.

Response: The introduction section was reviewed again according to the Reviewer’s comments. All changes are marked in red in the revised manuscript (Please see line: 69 – 78).

2.      Conclusion - The practical impact of the results obtained in this work should be better highlighted. The authors should also give an outlook on future research work.

Response: The relevant correction has been made according to the Reviewer’s comment. All changes are marked in red in the revised manuscript. All changes are marked in red in the revised manuscript (Please see line: 263 – 276).

Reviewer 3 Report

An interesting study regarding the potential hazards of endogenous fires in coal mines. The overall objectives of the study are clearly presented, the assessment methodology is comprehensive and sound, the results are very nice interpreted.

Overall, I would recommend to accept the paper.

Author Response

Reviewer #1:

An interesting study regarding the potential hazards of endogenous fires in coal mines. The overall objectives of the study are clearly presented, the assessment methodology is comprehensive and sound, the results are very nice interpreted.

Overall, I would recommend to accept the paper.

Response: Thank you for your positive opinion.